# SAC³: Reliable Hallucination Detection in Black-Box Language Models via Semantic-aware Cross-check Consistency

**Jiaxin Zhang[1], Zhuohang Li[2], Kamalika Das[1], Bradley Malin[2,3], Sricharan Kumar[1]**

[1]Intuit AI Research    [2]Vanderbilt University    [3]Vanderbilt University Medical Center

{jiaxin_zhang, kamalika_das, sricharan_kumar}@intuit.com
{zhuohang.li, b.malin}@vanderbilt.edu

## Abstract

Hallucination detection is a critical step toward understanding the trustworthiness of modern language models (LMs). To achieve this goal, we re-examine existing detection approaches based on the self-consistency of LMs and uncover two types of hallucinations resulting from 1) question-level and 2) model-level, which cannot be effectively identified through self-consistency check alone. Building upon this discovery, we propose a novel sampling-based method, i.e., semantic-aware cross-check consistency (SAC³) that expands on the principle of self-consistency checking. Our SAC³ approach incorporates additional mechanisms to detect both question-level and model-level hallucinations by leveraging advances including semantically equivalent question perturbation and cross-model response consistency checking. Through extensive and systematic empirical analysis, we demonstrate that SAC³ outperforms the state of the art in detecting both non-factual and factual statements across multiple question-answering and open-domain generation benchmarks.[1]

## 1 Introduction

Large-scale pre-trained language models (LMs) have demonstrated exceptional adaptability across a diverse array of natural language tasks that require generating open-ended responses based on user prompt comprehension (Zhao et al., 2023). However, prominent LMs like GPT (Brown et al., 2020) and PaLM (Chowdhery et al., 2022), often exhibit a tendency to produce exceedingly confident, yet erroneous, assertions commonly referred to as *hallucinations*. This phenomenon significantly impedes their applicability in domains where factual accuracy is of utmost importance.

Hallucinations can be detected through the assistance of metrics that capture the uncertainty about the output sequences. However, these metrics require access to token-level log probabilities, which are not available in commercial black-box LMs like ChatGPT or Bard that only offer limited API access. To overcome this limitation, recent studies explore sampling-based approaches for approximating uncertainty estimation (Lin et al., 2023) through establishing a connection between confidence and self-consistency (Manakul et al., 2023; Mündler et al., 2023). The underlying premise of this principle is that LMs are more inclined to generate consistent responses when high probabilities are assigned to tokens in the answer, which, in turn, implies a level of factuality. In contrast, inconsistent responses are more likely to contain hallucinations. To operationalize this concept, current approaches are designed to sample multiple responses from the LMs for a given question and then compose a hallucination score for each sentence based on the level of response consistency.

**Key Observations.** In this work, we investigate the relationship between the self-consistency of LMs and the occurrence of hallucinations in a diverse range of tasks. Our investigation indicates that while self-inconsistency in LMs often coincides with hallucination, self-consistency does not necessarily guarantee factual answers, as shown in Figure 1. Our findings challenge the notion that self-consistency alone can serve as a reliable indicator of veracity, as it is demonstrated that LMs can exhibit various tiers of hallucination that elude detection through self-consistency checks. One such tier is *question-level* hallucination, where LMs consistently generate incorrect answers in response to specific questions (e.g., Table 1). We reveal that by reformulating the questions, it is possible to mitigate such instances of hallucinations. Additionally, our work further reveals the existence of *model-level* hallucinations, whereby different LMs show discrepancies in their propensity for halluci-

---

[1]All resources are available at https://github.com/intuit/sac3.

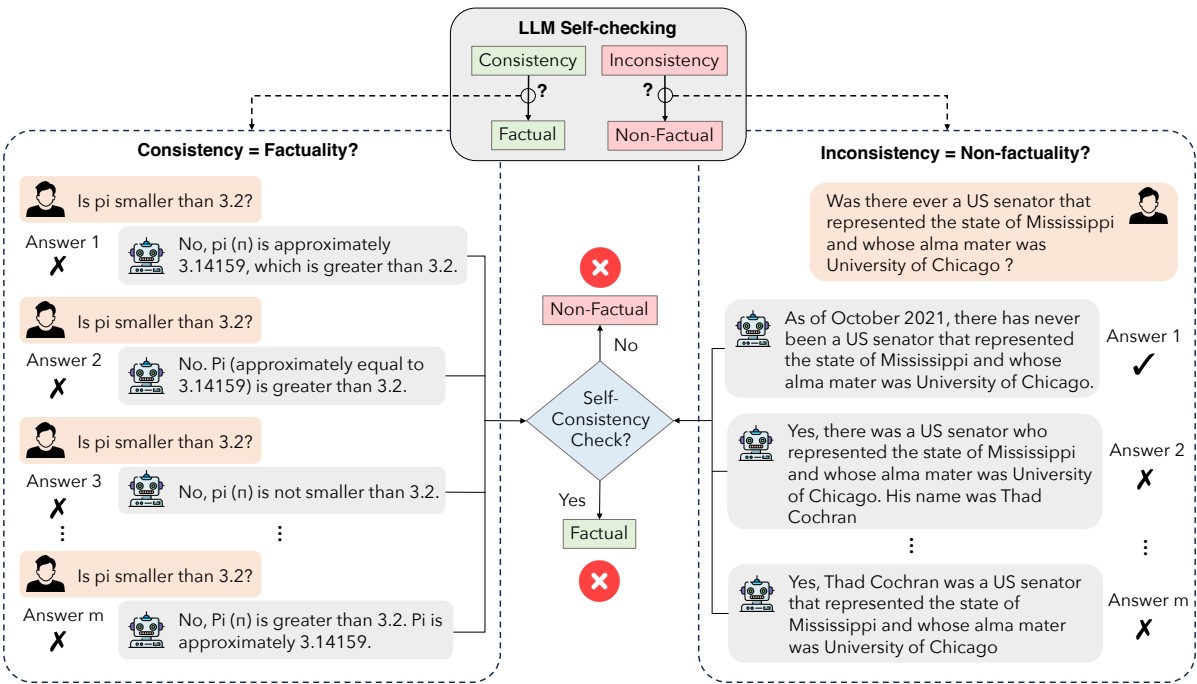

Figure 1: Key observation: solely checking the self-consistency of LLMs is not sufficient for deciding factuality. Left: generated responses to the same question may be consistent but non-factual. Right: generated responses may be inconsistent with the original answer that is factually correct.

nation. Surprisingly, we even observe cases where smaller LMs are capable of correctly answering questions for which larger LMs hallucinate. Together, these findings accentuate the need to consider model-specific characteristics when assessing the occurrence of hallucinations.

**Proposed Approach.** Motivated by these observations, we introduce SAC$^3$, a new sampling-based approach utilizing semantic-aware cross-check consistency to improve the detection of hallucinations in black-box LMs. An overview of our approach is provided in Fig. 2. To address question-level hallucination, we introduce a mechanism that perturbs semantically equivalent questions to evaluate the consistency of LMs' responses across variants of the same question. By examining the generated answers to these perturbed questions, we are able to identify cases where the LM consistently provides incorrect responses to a specific question, which is indicative of a question-level hallucination. Furthermore, we address model-level hallucination by introducing cross-model response consistency checking, which involves comparing the responses of different LMs to the same set of questions. By identifying discrepancies between the responses of different models, we are able to pinpoint cases where certain models exhibit hallucinations while

others provide correct answers. Integrating these cross-checking extensions into our approach significantly improves its ability to detect hallucinations that go beyond self-consistency, thereby providing a more comprehensive assessment of the presence of question-level and model-level hallucinations.

**Effectiveness of SAC$^3$.** We demonstrate the effectiveness of SAC$^3$ in multiple classification and open-domain generation QA datasets. In the classification QA tasks, our approach achieves a high AU-ROC score of 99.4% and 97.0% respectively, which significantly outperforms the self-consistency baseline. In the case of open-domain generation QA tasks, our approach achieves an AUROC score of 88.0% and 77.2%, which substantially improves upon the self-consistency baseline by a large margin (+13.8%, and +6.7% respectively).

## 2 Related Work

**Hallucination in LMs.** The issue of hallucination in language models (LMs) has gained significant attention due to its negative impact on performance and the risks it introduces in various natural language processing (NLP) tasks, such as machine translation (Zhou et al., 2020), summarization (Cao et al., 2022), dialogue generation (Das et al., 2023), and question answering (Zhang et al., 2023a; Zheng

| Method | Input | Model | Output | Consistency | Factuality |
|---|---|---|---|---|---|
| Self-consistency | Q: "Is 3691 a prime number?" | Target | "No, 3691 is not a prime number as it is divisible by 7 and 13." | ✓ | ✗ |
| | Q: "Is 3691 a prime number?" | Target | "No, 3691 is not a prime number as it is divisible by 7 and 13." | | |
| | Q: "Is 3691 a prime number?" | Target | "No, 3691 is not a prime number. It can be divided evenly by 7 and 19." | | |
| Cross-Q consistency | Q1: "Is the number 3691 prime?" | Target | "Yes, the number 3691 is prime." | ✗ | ✓ |
| | Q2: "Are the only factors of 3691 itself and one?" | Target | "Yes, the only factors of 3691 are itself and one." | | |
| | Q3: "Can 3691 only be divided by 1 and 3691?" | Target | "Yes, 3691 is a prime number and can only be divided by 1 and 3691." | | |
| Cross-M consistency | Q: "Is 3691 a prime number?" | Verifier | "Yes, 3691 is a prime number." | ✗ | ✓ |
| Cross-M & Q consistency | Q1: "Is the number 3691 prime?" | Verifier | "Yes." | ✗ | ✓ |
| | Q2: "Are the only factors of 3691 itself and one?" | Verifier | "Yes, the only factors of 3691 are 1 and itself." | | |
| | Q3: "Can 3691 only be divided by 1 and 3691?" | Verifier | "Yes, 3691 can only be divided by 1 and 3691." | | |

Table 1: An illustrative example of self-consistency, cross-question consistency, and cross-model consistency check. The original question and answer are "Is 3691 a prime number?" and "No, 3691 is not a prime number. It is divisible by 7 and 13", respectively. Each row presents a set of sampled QA pairs along with its consistency regarding the original answer, and the predicted factuality of the original answer.

et al., 2023b; Dhuliawala et al., 2023). Recent survey (Ji et al., 2023; Zhang et al., 2023c; Ye et al., 2023) and evaluation benchmarks (Liu et al., 2021; Li et al., 2023a; Yang et al., 2023) have highlighted the importance of addressing this issue. Previous research has explored hallucination evaluation using confidence-based approaches (Xiao and Wang, 2021; Varshney et al., 2023; Chen and Mueller, 2023) that require access to token-level log probability (Kuhn et al., 2023; Cole et al., 2023) or supervised tuning (Agrawal et al., 2023; Li et al., 2023b) that relies on internal states of the LM. However, these methods may not be applicable when only API access to the LM is available (Agrawal et al., 2023). Another approach involves retrieving knowledge from external databases to tackle hallucinations (Ji et al., 2022; Zheng et al., 2023a; Peng et al., 2023; Zhang et al., 2023b). In contrast to these studies, our work focuses on detecting hallucinations in open-domain QA tasks using black-box LMs, without relying on external resources.

**Consistency Evaluation of LMs.** An essential characteristic of logically valid intelligent systems is self-consistency, which entails that no two statements provided by the system contradict each other. Self-consistency is defined by Elazar et al. (2021) as the invariance of an LM's responses across various types of semantics-preserving prompt transformations. This definition is further enriched by multiple other consistency categories proposed by Jang et al. (2022). Wang et al. (2022) demonstrates that self-consistency can significantly enhance the chain of thought reasoning in LMs. Without self-

consistency, it becomes challenging to regard LMs as reliable or trustworthy systems. Recent studies employ self-consistency to detect hallucinations based on pretrained LMs (Manakul et al., 2023) and instruction-tuned LMs (Mündler et al., 2023). Although these methods exhibit promising accuracy on several specific tasks, potential failures (Chen et al., 2023) of self-consistency are overlooked in the current settings, as existing LMs frequently provide inconsistent responses to questions (Mitchell et al., 2022) and factual knowledge inquiries (Elazar et al., 2021; Tam et al., 2023; Gekhman et al., 2023). Our work addresses these concerns by introducing a cross-check consistency approach, aiming to bridge the gap between self-consistency and factual assessment.

## 3 Self-consistency Limitations in Factuality Assessment

The essential assumption of self-consistency in factuality assessment is that if the LM has the knowledge of the concept, responses sampled from its output distribution, should be similar and consistent; conversely, if the LM lacks corresponding knowledge, the sampled responses would contain hallucinated facts that are diverged and contradictory. Although this assumption may seem reasonable, it does not always hold in practice (more details are provided in the Appendix A). Specifically, we argue that solely checking the LM's self-consistency is insufficient for detecting hallucination or verifying factuality under the following two circumstances:

*1. LMs may produce consistently hallucinated facts.* We observe that for certain questions, LMs may output consistently wrong answers. For instance, as shown in Fig. 1, when prompted with the question "Is pi smaller than 3.2?", ChatGPT consistently generates incorrect answers. In this case, where the generated responses are consistent but non-factual, solely relying on self-consistency checking of a single model would yield false negative hallucination detection results.

*2. Even in cases when LMs generate factual statements in their original response, the stochastic sampled responses may lack veracity.* For example, the original answer (Answer 1) of ChatGPT under zero temperature is correct regarding the senator search question as shown in Fig. 1. However, when sampled with a higher temperature, ChatGPT generates multiple incorrect responses (Answer 2 and Answer $m$). In this scenario, where the sampled responses are inconsistent and disagree with the original response which itself is factually correct, methods that rely solely on model self-checking would produce false positives.

In summary, although the inconsistency of sampled responses has been empirically demonstrated to be correlated with hallucinated facts on certain tasks, in general, self-consistency is neither necessary nor sufficient to verify the veracity of large LMs' statements. Therefore, methods based solely on self-consistency checking may not be able to accurately detect hallucinations in complex QA and open-domain generation tasks, which motivates us to design a more reliable and robust factuality assessment method that extends this idea.

## 4   SAC³ : Semantic-Aware Cross-check Consistency

This section describes the proposed semantic-aware cross-check consistency approach, a high-level overview of which is provided in Fig. 1. Additionally, an illustrative example of each component is presented in Table 1. Here, we walk through each component in detail.

### 4.1   Stage 1: Question-level Cross-checking via Semantically Equivalent Perturbations

Contrary to existing techniques that assess semantic equivalence through entailment or paraphrasing, our approach involves rephrasing the input query by generating alternative inputs that preserve semantic equivalence, i.e., semantically equivalent input perturbation. To achieve this, we leverage advances in LLM prompting. Starting with a queried input $x_0$, we acquire a set of $k$ semantically equivalent inputs $\{x_1, x_2, ..., x_k\}$ through the prompt: "For the question [QUERIED QUESTION], provide $k$ semantically equivalent questions".

To ensure the quality of the generated inputs in this step, we further double-check the semantic equivalence between the generated inputs $\{x_1, x_2, ..., x_k\}$ and the queried input $x_0$ in a pair-wise manner using the prompt "Are the following two inputs semantically equivalent? [QUERIED INPUT] [GENERATED INPUT]" and filtering out the inputs that do not share the same semantic meaning as the original input. The complete prompt templates used in this work are provided in the Appendix B.

### 4.2   Stage 2: Model-level Cross-check with Additional Verifier LM

Let $s_0$ denote the original response from a target LM $\mathcal{T}$ based on a given query $x_0$. Our objective is to detect whether $s_0$ is hallucinated by sampling responses from the predictive distribution of $\mathcal{T}$. To avoid model-level hallucination, we introduce an additional verifier LM denoted as $\mathcal{V}$ for model-level cross-checking. We define the responses from both models as:

$$s_{\mathcal{T}_j} = \mathcal{T}(x_j), \ s_{\mathcal{V}_j} = \mathcal{V}(x_j), \ j = 1, ..., k, \quad (1)$$

where $k$ is the length of the generated semantically equivalent inputs $\{x_1, x_2, ..., x_k\}$ in stage 1. To assess the factuality of $x_0$, the self-checking mechanism operates by drawing a set of $n_s$ stochastic response samples from the target LM: $\mathcal{S}_{\mathcal{T}_0} = \{s_{\mathcal{T}_0}^1, s_{\mathcal{T}_0}^2, ..., s_{\mathcal{T}_0}^{n_s}\}$. Similarly, we can apply the same self-checking mechanism to the verifier LM to generate another set of $n_m$ responses: $\mathcal{S}_{\mathcal{V}_0} = \{s_{\mathcal{V}_0}^1, s_{\mathcal{V}_0}^2, ..., s_{\mathcal{V}_0}^{n_m}\}$. To perform question-level cross-check, for each perturbed input $x_k$, we generate $n_q$ sampled response sequences $\mathcal{S}_{\mathcal{T}_k} = \{s_{\mathcal{T}_k}^1, s_{\mathcal{T}_k}^2, ..., s_{\mathcal{T}_k}^{n_q}\}$ from the target LM $\mathcal{T}$ and $n_{qm}$ sampled responses $\mathcal{S}_{\mathcal{V}_k} = \{s_{\mathcal{V}_k}^1, s_{\mathcal{V}_k}^2, ..., s_{\mathcal{V}_k}^{n_{qm}}\}$ from the verifier LM $\mathcal{V}$.

Finally, we collect the total sampled sets $\mathcal{S} = \{\mathcal{S}_{\mathcal{T}_0}, \mathcal{S}_{\mathcal{V}_0}, \mathcal{S}_{\mathcal{T}_k}, \mathcal{S}_{\mathcal{V}_k}\}$ by combining all samples drawn from self-checking and cross-checking, which will be used next for calculating a consistency score.

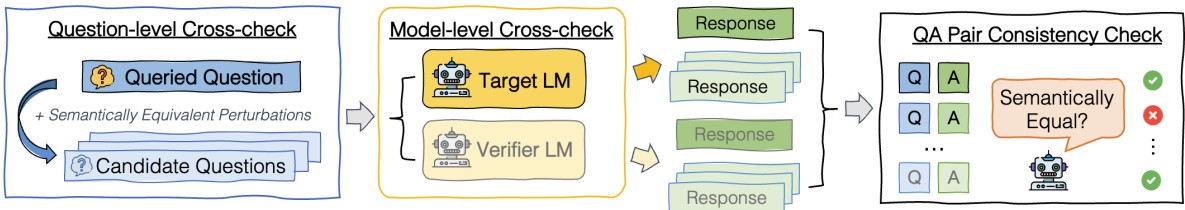

Figure 2: Overview of the proposed semantic-aware cross-check consistency ($\text{SAC}^3$) method.

## 4.3 Stage 3: Consistency Score Calculation

This stage uses the generated sample sets in all previous stages to calculate a numerical consistency score that captures the question-level and model-level cross-checking paradigm.

### 4.3.1 Semantic-aware Consistency Check of QA Pairs

Most of the existing works mainly focus on examining the consistency of LM outputs while ignoring the effect of the inputs. However, in QA tasks, it is important to consider both inputs and outputs when measuring semantic equivalence, as the same question can be rephrased in many ways. Although the answers to these questions (e.g., "no" and "yes") may not be lexically equivalent, the QA pairs as a whole can be semantically equivalent. In light of this, we propose to check the semantic consistency of the QA pairs instead of the answer only.

### 4.3.2 Self-checking Consistency ($\text{SC}^2$) Score

Let $\mathcal{C}(\cdot, \cdot)$ denote a semantic equivalence checking operator that takes two QA pairs as inputs. The operator $\mathcal{C}$ returns "Yes" if the two QA pairs are semantically equivalent, and "No" otherwise. This operator should be reflexive, symmetric, and transitive. We implement the checking operator using an LM by leveraging the prompt: "Are the following two Question-Answering (QA) pairs semantically equivalent? [QA PAIR 1] [QA PAIR 2]". We then map the best guess to a numerical semantic equivalent score: {"Yes" $\rightarrow$ 0.0, "No" $\rightarrow$ 1.0}. We use $\mathcal{P}_0 = (\boldsymbol{x}_0, \boldsymbol{s}_0)$ to denote the original QA pair. The self-checking score $\mathcal{Z}_{\text{SC}^2}$ of the target LM $\mathcal{T}$ can be calculated by

$$\mathcal{Z}_{\text{SC}^2} = \frac{1}{n_s} \sum_{i=1}^{n_s} \mathcal{C}(\mathcal{P}_0, \mathcal{P}_{\mathcal{S}_{\mathcal{T}_0}}^i), \quad (2)$$

where $\mathcal{P}_{\mathcal{S}_{\mathcal{T}_0}} = \{(\boldsymbol{x}_0, \boldsymbol{s}_{\mathcal{T}_0}^1), ..., (\boldsymbol{x}_0, \boldsymbol{s}_{\mathcal{T}_0}^{n_s})\}$ represents the QA pairs generated in the self-checking scenario.

### 4.3.3 Question-level Consistency ($\text{SAC}^3\text{-Q}$) Score

Besides self-checking the original question $\boldsymbol{x}_0$, $\text{SAC}^3$ further assesses cross-check consistency of perturbed questions $\{\boldsymbol{x}_1, \boldsymbol{x}_2, ..., \boldsymbol{x}_k\}$. The corresponding QA pairs compose a two-dimensional matrix, where each row corresponds to a perturbed question ($k$ in total), and each column corresponds to a sampled response ($n_q$ in total):

$$\mathcal{P}_{\mathcal{S}_{\mathcal{T}_j}}^i = \begin{bmatrix} (\boldsymbol{x}_1, \mathcal{S}_{\mathcal{T}_1}^1) & ... & (\boldsymbol{x}_1, \mathcal{S}_{\mathcal{T}_1}^{n_q}) \\ ... & ... & ... \\ (\boldsymbol{x}_k, \mathcal{S}_{\mathcal{T}_k}^1) & ... & (\boldsymbol{x}_k, \mathcal{S}_{\mathcal{T}_k}^{n_q}) \end{bmatrix}. \quad (3)$$

Therefore, the question-level cross-checking consistency score $\mathcal{Z}_{\text{SAC}^3\text{-Q}}$ can be obtained by

$$\mathcal{Z}_{\text{SAC}^3\text{-Q}} = \frac{1}{n_q \cdot k} \sum_{i=1}^{n_q} \sum_{j=1}^{k} \mathcal{C}(\mathcal{P}_0, \mathcal{P}_{\mathcal{S}_{\mathcal{T}_j}}^i). \quad (4)$$

### 4.3.4 Model-level Consistency ($\text{SAC}^3\text{-M}$ & $\text{SAC}^3\text{-QM}$) Score

In addition to the question-level score, a model-level cross-check score is calculated by performing cross-model checking and cross-question checking using the verifier LM $\mathcal{V}$. Specifically, for the original question $\boldsymbol{x}_0$, the model-level cross-checking consistency score $\mathcal{Z}_{\text{SAC}^3\text{-M}}$ is computed by

$$\mathcal{Z}_{\text{SAC}^3\text{-M}} = \frac{1}{n_m} \sum_{i=1}^{n_m} \mathcal{C}(\mathcal{P}_0, \mathcal{P}_{\mathcal{S}_{\mathcal{V}_0}}^i), \quad (5)$$

where $\mathcal{P}_{\mathcal{S}_{\mathcal{V}_0}} = \{(\boldsymbol{x}_0, \boldsymbol{s}_{\mathcal{V}_0}^1), ..., (\boldsymbol{x}_0, \boldsymbol{s}_{\mathcal{V}_0}^{n_m})\}$ is the QA pairs generated by the verified LM $\mathcal{V}$.

The cross-question consistency score on the verifier LM is computed on the QA pairs produced by $\mathcal{V}$:

$$\mathcal{P}_{\mathcal{S}_{\mathcal{V}_j}}^i = \begin{bmatrix} (\boldsymbol{x}_1, \mathcal{S}_{\mathcal{V}_1}^1) & ... & (\boldsymbol{x}_1, \mathcal{S}_{\mathcal{V}_1}^{n_{qm}}) \\ ... & ... & ... \\ (\boldsymbol{x}_k, \mathcal{S}_{\mathcal{V}_k}^1) & ... & (\boldsymbol{x}_k, \mathcal{S}_{\mathcal{V}_k}^{n_{qm}}) \end{bmatrix}. \quad (6)$$

The cross-model cross-question consistency score

can thus be obtained through

$$\mathcal{Z}_{\text{SAC}^3\text{-QM}} = \frac{1}{n_{qm} \cdot k} \sum_{i=1}^{n_{qm}} \sum_{j=1}^{k} \mathcal{C}(\mathcal{P}_0, \mathcal{P}_{\mathcal{S}_{\mathcal{V}_j}}^i). \quad (7)$$

### 4.3.5 Final Score and Model Confidence

The different variants of SAC$^3$ capture different aspects of the uncertainty about the original response and should complement each other. We thus consider a combination of all variants including SAC$^3$-Q, SAC$^3$-M, SAC$^3$-QM as the final score:

$$\mathcal{Z}_{\text{SAC}^3\text{-all}} = \mathcal{Z}_{\text{SAC}^3\text{-Q}} + \lambda(\mathcal{Z}_{\text{SAC}^3\text{-M}} + \mathcal{Z}_{\text{SAC}^3\text{-QM}}), \quad (8)$$

where $\lambda$ is a weight factor for the verifier LM. Unless mentioned otherwise, we use $\lambda = 1$ by default in our experiments. In practice, as the computation of each component is independent, they can be computed in parallel to reduce latency. The detection prediction is made by comparing the final score with a preset threshold. In addition to the computed score, we also ask the target LM to generate a verbalized confidence score (Tian et al., 2023) along with its prediction when checking the semantic equivalence of QA pairs. More discussions are offered in the Appendix C.2.

## 5 Data and Annotation

We evaluate our hallucination detection approach on two categories of QA tasks, namely, classification QA and generation QA, with each category containing two datasets. Following prior work (Zhang et al., 2023a), we use the following two binary classification datasets for evaluation on the classification QA task:

- **Prime number**: this dataset contains 500 questions that query the primality of a randomly chosen prime number between 1,000 and 20,000, where the factual answer is always "Yes". The synthesized hallucinated answers are "No, it is not a prime number".

- **Senator search**: the dataset consists of 500 questions that follow the following template: "Was there ever a US senator that represented the state of [US STATE NAME] and whose alma mater was [US COLLEGE NAME]?". The factual answer is always "No". We also generate hallucinated answers: "Yes, there was a US senator that represented the state of [US STATE NAME] and whose alma mater was [US COLLEGE NAME].".

As for the generation QA tasks, we take questions from the following two open-domain QA datasets and generate answers using LLMs. Then we manually annotate the factuality of the answers following previous work (Li et al., 2023a).

- **HotpotQA-halu**: We randomly sample 250 examples from the training set of HotpotQA (Yang et al., 2018) and generate hallucinated answers drawn from gpt-3.5-turbo. Then we manually annotate the answers by comparing the ground truth and knowledge.

- **NQ-open-halu**: Natural Questions (NQ)-open (Lee et al., 2019) is a more challenging open domain QA benchmark (Kwiatkowski et al., 2019). We use the same setting as HotpotQA-halu to create a small-scale dataset that consists of 250 non-factual and factual examples with manual annotations.

Please find more relevant details about data annotations in the Appendix C.3.

## 6 Experiments

### 6.1 Experimental Setup

**Evaluation Models.** We use gpt-3.5-turbo from OpenAI as the target LM for our experiment. The verifier LM is chosen from the following two models: (1) Falcon-7b-instruct (Almazrouei et al., 2023): an open-source causal decoder-only model built by TII that is trained on 1,500B tokens of RefinedWeb (Penedo et al., 2023) and further enhanced using the curated corpora; and (2) Guanaco-33b: an open-source instruction-following models through QLoRA (Dettmers et al., 2023) tuning of LLaMA (Touvron et al., 2023) base model on the OASST1 dataset.

**Implementation Details.** The evaluation is conducted using Azure OpenAI API. When performing semantic perturbations and consistency checking, we set the temperature to 0.0 to get deterministic high-quality outputs. Given a specific input query, we generate $k = 10$ semantically equivalent inputs using the prompt described in Section 4.1. For the self-checking-based method SC$^2$, we follow prior work (Manakul et al., 2023) to set the temperature to 1.0 and generate $n_s = 10$ stochastic samples. For SAC$^3$-Q and SAC$^3$-QM, we set $n_q = n_{qm} = 1$ to reduce computational cost. To further reduce the inference cost, we set $n_m = 1$ by default and

combine SAC³-M with SAC³-QM to report the model-level results. We use hallucination detection accuracy and area under the ROC curve (AUROC) to evaluate the performance. In addition to the estimated hallucination score, we also show the verbalized probabilities ([Tian et al., 2023](#)) from the target LM for comparison. We execute all experiments on 8 NVIDIA V100 32G GPUs.

| Method | Prime number | | Senator search | |
|---|---|---|---|---|
| | Score | Confidence | Score | Confidence |
| SC² (gpt-3.5-turbo) | 65.9 | 67.5 | 56.1 | 53.1 |
| SAC³-Q (gpt-3.5-turbo) | **99.4** | **99.7** | **99.7** | **99.7** |

Table 2: AUROC on classification QA tasks with 50% hallucinated samples and 50% factual samples.

## 6.2 Evaluation Results

### 6.2.1 Classification QA

**Balanced Dataset.** We first experiment on balanced datasets with 50% hallucinated samples and 50% factual samples. Table 2 compares the detection performance of SC² and SAC³-Q in terms of AUROC and verbalized confidence score. We observe that self-checking (SC²) performs poorly on both datasets, with a low AUROC of 65.9% and 56.1%, respectively. Our question-level cross-checking (SAC³-Q) significantly outperforms the SC² baseline achieving > 99% AUROC on both datasets and is in line with the verbalized confidence score, confirming the effectiveness of cross-checking.

**Unbalanced Dataset.** We further evaluate our method in a more challenging scenario where the dataset only contains hallucinated samples. Table 3 presents the accuracy of detecting hallucinated samples using a preset threshold of 0.5. In this case, the performance of self-check drops significantly to 48.2% and 29.6% respectively. SAC³-Q still outperforms SC² by a large margin. The model-level cross-check with verifier LMs performs well in the prime number dataset but fails to accurately detect hallucination in the senator search dataset. This is because both verifier LMs refuse to answer a large portion of the questions on this dataset due to a lack of sufficient information. By combining the target LM, SAC³-all with Guanaco-33b achieves the highest detection accuracy compared to other baselines SC² (+51.2%), SAC³-Q (+6.2%), and SAC³-QM (+ 5.0%).

| Method | Prime number | | Senator search | |
|---|---|---|---|---|
| | Score | Confidence | Score | Confidence |
| SC² (gpt-3.5-turbo) | 48.2 | 51.0 | 29.6 | 30.6 |
| SAC³-Q (gpt-3.5-turbo) | 93.2 | 96.4 | **97.0** | **97.4** |
| SAC³-QM (Falcon-7b) | 89.8 | 91.2 | 21.0 | 21.8 |
| SAC³-QM (Guanaco-33b) | 94.4 | 96.3 | 45.6 | 46.2 |
| SAC³-all (Falcon-7b) | 97.8 | 98.0 | 84.6 | 85.6 |
| SAC³-all (Guanaco-33b) | **99.4** | **99.4** | 85.6 | 86.7 |

Table 3: Accuracy on classification QA tasks with 100% hallucinated samples (with the threshold set to 0.5).

**Impact of Threshold.** Since the choice of threshold has a significant impact on the detection accuracy in the case where the dataset only contains positive (hallucinated) samples, we further experiment with different thresholds and present the results in Fig. 3. We observe that SAC³-Q and SAC³-all with Guanaco-33b are more robust against large threshold values and outperform SC² in most cases.

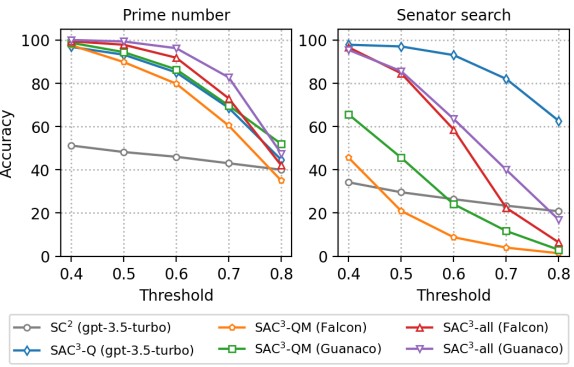

Figure 3: Impact of threshold on detection accuracy.

**Why Does Self-checking Fail?** To further understand why self-checking methods fail to detect some hallucinated responses, we visualize the distribution of consistency scores using histogram plots in Fig. 4. We observe that for SC², a significant portion of hallucinated samples received highly consistent predictions. In other words, the target LM made consistently wrong predictions due to a lack of question and model diversity, which aligns with our analysis in Section 3. On the other hand, benefiting from the semantically equivalent question perturbation, SAC³-Q's scores are more spread out in the inconsistent region, which helps to improve the effectiveness of detecting hallucinations by choosing a proper threshold.

### 6.2.2 Open-domain Generation QA

Compared to the classification QA tasks, detecting hallucinations in open-domain generation QA

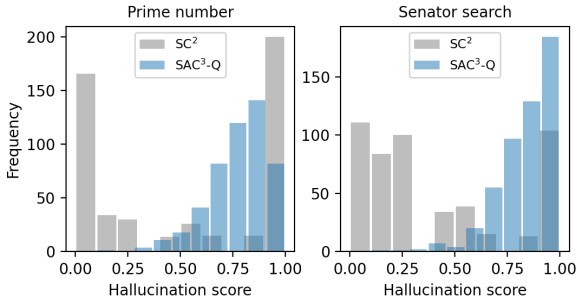

Figure 4: Histogram of hallucination score.

tasks is more challenging. As shown in Table 4, $\text{SAC}^3\text{-Q}$ exhibits better AUROC than $\text{SC}^2$ (+7%) in both datasets. Compared to $\text{SAC}^3\text{-Q}$, $\text{SAC}^3\text{-QM}$ shows 6.7% improvement in the HotpotQA-halu dataset but is slightly worse in the NQ open dataset. $\text{SAC}^3\text{-all}$ leverages the advantages of question-level and model-level cross-checking and achieves consistently good performance in both datasets.

| Method | HotpotQA-halu | | NQ-open-halu | |
|---|---|---|---|---|
| | Score | Confidence | Score | Confidence |
| $\text{SC}^2$ (gpt-3.5-turbo) | 74.2 | 77.0 | 70.5 | 72.7 |
| $\text{SAC}^3\text{-Q}$ (gpt-3.5-turbo) | 81.3 | 81.4 | **77.2** | **78.5** |
| $\text{SAC}^3\text{-QM}$ (Falcon-7b) | 83.0 | 79.5 | 67.5 | 62.0 |
| $\text{SAC}^3\text{-QM}$ (Guanaco-33b) | **88.0** | 85.2 | 72.7 | 72.7 |
| $\text{SAC}^3\text{-all}$ (Falcon-7b) | 84.5 | 84.5 | 77.1 | 77.2 |
| $\text{SAC}^3\text{-all}$ (Guanaco-33b) | 87.0 | **86.8** | **77.2** | 77.8 |

Table 4: AUROC on open-domain generation QA tasks.

**Effect of Verifier LM Weight.** In our previous experiments, we assigned equal importance to the consistency score computed by the target and verifier LM. However, typically, the target LM and the verifier LM have different architectures and scales such that the user may have different levels of trust in their output truthfulness. This difference in trust can be incorporated by introducing a weight $\lambda$ to the consistency score produced by the verifier LM. For instance, if the goal is to detect hallucination in a specialized domain and the verifier LM is a domain-specific model developed for this domain, we can assign a large weight to its scores (e.g., $\lambda > 1.0$). In the general case, where the verifier LM is a small-sized open-source model, we can apply a small weight value (e.g., $\lambda < 1.0$) to discount the influence of the verifier LM in the final score. Fig. 5 visualizes the effect of various weight factors on the generation QA datasets. We observe that $\text{SAC}^3\text{-all}$ with a higher weight would result in a larger advantage over $\text{SAC}^3\text{-Q}$ in

the HotpotQA-halu task, where the verifier LM outperforms the target LM. On the contrary, in the NQ open task where the target LM shows competitive performance, a smaller weight would yield better results.

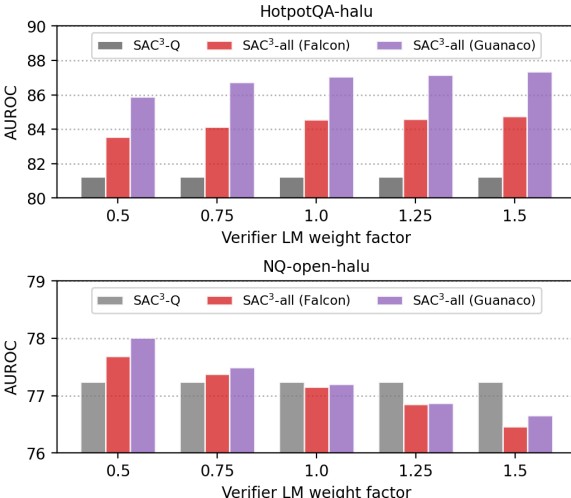

Figure 5: Effect of verifier LM weight on AUROC.

**Effect of the Number of Perturbed Questions.** The performance of sampling-based methods is expected to improve as the sample size increases, at the cost of higher latency and computational cost. In Fig. 6, we study this trade-off by varying the number of perturbed questions $k$ from 2 to 10. We observe that the performance of $\text{SAC}^3$ increases as more question samples are used but the performance gain gradually diminishes after using more than 5 question samples. This suggests that in practice we could use 2-4 question samples to achieve reasonably good performance at a low computational cost.

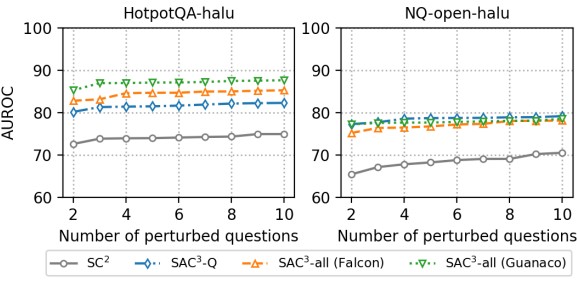

Figure 6: Performance of $\text{SAC}^3$ with varying number of perturbed questions.

**Effect of the Model Type.** Our proposed $\text{SAC}^3$ framework does not restrict the type of LLM employed and can be naturally extended to various

types of target LLMs. To verify this, in addition to GPT-3.5 (`gpt-3.5-turbo`), we conduct experiments using GPT-4 (`gpt-4`) and PaLM 2 (`chat-bison`) on the considered four datasets in the setting of the balanced dataset. The experimental results of comparing the proposed $\mathtt{SAC}^3$-$\mathtt{Q}$ with the $\mathtt{SC}^2$ baseline are summarized in Table 5. We observe that the proposed $\mathtt{SAC}^3$-$\mathtt{Q}$ consistently outperforms the $\mathtt{SC}^2$ baseline across all LLM variants.

| Method | Prime Number | Senator Search | HotpotQA -halu | NQ-open -halu |
|---|---|---|---|---|
| $\mathtt{SC}^2$ (`gpt-3.5-turbo`) | 48.2 | 29.6 | 74.2 | 70.5 |
| $\mathtt{SC}^2$ (`gpt-4`) | 38.3 | 18.4 | 79.7 | 76.3 |
| $\mathtt{SC}^2$ (`chat-bison`) | 26.9 | 19.2 | 75.8 | 67.9 |
| $\mathtt{SAC}^3$-$\mathtt{Q}$ (`gpt-3.5-turbo`) | **93.2** | **97.0** | 81.3 | 77.2 |
| $\mathtt{SAC}^3$-$\mathtt{Q}$ (`gpt-4`) | 91.1 | 61.6 | **87.2** | **82.9** |
| $\mathtt{SAC}^3$-$\mathtt{Q}$ (`chat-bison`) | 90.3 | 66.3 | 82.8 | 72.7 |

Table 5: Accuracy of different LLMs (GPT-3.5, GPT-4, and PaLM 2) on classification and generation QA tasks.

**Computational Cost.** We monitor the computational cost of our approach based on the number of model evaluations consumed by OpenAI API and open-source LLMs inference. Assuming the number of samples equals the number of perturbed questions, i.e., $n_s = n_m = n_q = n_{qm}$, the cost of $\mathtt{SC}^2$ is $n_s$ API calls, and our $\mathtt{SAC}^3$-$\mathtt{all}$ needs $n_s$ target LM calls plus $2 \times n_s$ verifier LM calls. Beyond the model evaluations, $\mathtt{SAC}^3$ may have additional costs from question perturbations and semantic equivalence checking via prompting. Additional discussions can be found in the Appendix C.1.

## 7 Conclusion and Discussion

We investigate the relationship between the self-consistency of LM and the factuality of the response and propose $\mathtt{SAC}^3$ as a robust hallucination detection approach for black-box LMs. Through extensive empirical analysis, our work highlights several findings. *First*, self-consistency checking alone is insufficient to effectively detect question-level and model-level hallucinations, where LMs generate consistently wrong responses to certain questions. *Second*, cross-checking between semantically equivalent questions can reduce the occurrence of persistent hallucinations, potentially by reducing question-level ambiguity. *Third*, there exists a model-level disparity in hallucinations, which we attribute to the inherent differences in LM capabilities originating from different training procedures and data. Thus, the verifier LM can be selected

according to specific tasks to maximize detection accuracy. We believe that our work is an important step towards building reliable LLMs.

## Ethics Statement

This paper studies hallucination detection in LMs, which has significant broader impacts in the field of natural language processing (NLP) and helps to address ethical considerations regarding trustworthiness and reliability. The research outcome may contribute to the development of more accurate and reliable LMs by mitigating the risks of misinformation and biased outputs and promoting accountability and trust in AI systems.

## Limintations

Our current experiments focus on the question-answering setting. Further research is needed to assess the generalizability of the proposed framework and the accuracy of semantic equivalence checks on more complex tasks such as conversational or dialogue-based prompting. Additionally, it would be interesting to investigate the efficiency-utility trade-off: we expect increasing sample sizes to improve detection accuracy but may introduce additional cost and latency. Speeding up the implementation through parallelization is also worth exploring.

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

## A Factuality Assessment of LLMs

This section first introduces the background of LM factuality assessment and reviews the key components of existing black-box factuality assessment approaches.

### A.1 Background

Existing studies (Manakul et al., 2023) showed that the factuality of LMs' statements can be assessed by capturing the output uncertainty. If permitted with complete white-box access, hallucinations of LMs can be detected through uncertainty metrics such as entropy, which models the uncertainty of the output distribution as $\mathcal{H}(\boldsymbol{s}) = -\sum_{i=1}^{N} p(s_i|s_{<i}) \log p(s_i|s_{<i})$ for a sequence of $N$ tokens $\boldsymbol{s} = \{s_i\}_{i=1}^{N}$. The insight is that factual sentences are likely to be composed of high-probability tokens and thus should have less entropy, whereas non-factual statements are likely to be associated with lower probability and higher entropy.

Although these methods provide an accurate estimate of the LM's uncertainty, computation of such metrics involves token-level output probability distribution, which is typically not accessible for prominent LMs like ChatGPT that are only available through black-box interactions.

### A.2 Black-box Assessment via Checking Consistency in Sampled Responses

One natural way to extend the above white-box factuality assessment methods to restrictive black-box settings is to approximate the uncertainty metrics by sampling LM responses. Existing approaches (Manakul et al., 2023) mostly focus on assessing factuality through inspecting LM's *self-consistency*. The typical process of self-consistency check is as follows. First, for a given user-queried prompt $x$, let the LM generate a candidate response $\boldsymbol{s}$ with a low (default) temperature. Next, to assess the factuality of the candidate statements, a higher temperature value is used to stochastically sample a diverse set of supporting responses $\{\boldsymbol{s}^1, \boldsymbol{s}^2, ..., \boldsymbol{s}^m\}$ from the LM. Finally, a hallucination score $\mathcal{S} \in [0, 1]$ is calculated by measuring the consistency between the candidate response and the supporting responses: $\mathcal{S} \to 0$ if the supporting responses agree with the candidate response and $\mathcal{S} \to 1$ if they disagree. The consistency between two responses is typically measured by checking the textual entailment or lexical simi-

larity using an additional pre-trained NLU model such as BERT (Manakul et al., 2023; Kuhn et al., 2023).

## B Complete Prompts

The prompt templates that we used are provided in Table 7, specifically the question perturbations using semantically equivalent prompts and semantic equivalence checks between two QA pairs.

## C Additional Details and Discussions

### C.1 Computational Cost

High computational cost is a common research challenge faced in many areas of black-box LLM studies, including self-consistency (Wang et al., 2022) and self-check (Manakul et al., 2023). Compared with these existing works, our framework enables model-level parallel execution of LLM inferences (either target or verifier LM) in the form of parallel API calls, which helps to mitigate the overhead. Moreover, we may further employ more advanced prompt strategies for semantic consistency checking that condense the pairwise comparisons to a single inference call to substantially reduce the time complexity. Finally, our framework also offers the flexibility for the user to adjust the balance between computational cost and accuracy: users may choose to trade off accuracy for efficiency in scenarios where low overhead is more valued than high detection precision.

To achieve a more balanced trade-off, we primarily recommend two strategies: (1) Utilizing a smaller sample size, such as the number of question perturbations and self-evaluations. Our ablation experiments demonstrate that the performance of SAC³ improves with an increase in question samples, but the performance gain plateaus after exceeding five question samples. This indicates that using 2-4 question samples in practice could yield satisfactory results with minimal computational expense. (2) Implementing an optimized prompt strategy. With the original input question, we prompt the GPT model to generate multiple paraphrases in a single API call, i.e., $\mathcal{O}(1)$ for question paraphrasing. Although semantic consistency checking via a pairwise method can be slightly burdensome, we can devise an advanced prompt strategy to parallelize the pairwise consistency check, i.e., through one API call instead of multiple calls, effectively reducing the inference complexity from $\mathcal{O}(n)$ to $\mathcal{O}(1)$. These strategies empower users

| Method | Input | Model | Output | Semantic Consistency |
|--------|-------|-------|--------|----------------------|
| $\text{SC}^2$ | Q | Target | Yes, there was a US senator that represented the state of North Dakota and whose alma mater was MIT. His name was Quentin N. Burdick. | ✓ |
| | Q | Target | Yes, there was a US senator who represented the state of North Dakota and whose alma mater was MIT. His name was Quentin N. Burdick. | ✓ |
| $\text{SAC}^3\text{-Q}$ | Q1 | Target | As an AI language model, I do not have access to the most recent updates or news. However, as of my last update, there is no North Dakota US senator who graduated from MIT. | ✗ |
| | Q2 | Target | As an AI language model, I do not have access to real-time information or current events. However, as of my last update, I am not aware of any US senators from North Dakota who were also MIT graduates. | ✗ |
| $\text{SAC}^3\text{-M}$ | Q | Verifier | No, there has never been a US senator who represented the state of North Dakota and attended MIT. | ✗ |
| $\text{SAC}^3\text{-QM}$ | Q1 | Verifier | No, a North Dakota US senator did not have MIT listed as their alma mater. | ✗ |
| | Q2 | Verifier | There are no known US senators from North Dakota who are also MIT graduates. | ✗ |

Table 6: **Input Question (Q) and Answer (A)**: Q = "Was there ever a US senator that represented the state of North Dakota and whose alma mater was MIT? " A = "Yes, there was a US senator who represented the state of North Dakota and whose alma mater was MIT. His name was Quentin Burdick." **Semantically rephrased questions:** Q1 = "Did a North Dakota US senator have MIT listed as their alma mater? " Q2 = "Were there any US senators from North Dakota who were also MIT graduates?

| Objective | Template |
|-----------|----------|
| Semantic question perturbation | For the question [QUERIED QUESTION], provide $\{k\}$ semantically equivalent questions \n Are the following two inputs semantically equivalent? \n [QUERIED INPUT] \n [GENERATED INPUT] |
| Semantic equivalence check | Are the following two Question-Answering (QA) pairs semantically equivalent? Provide your best guess and the probability that it is correct (0.0 to 1.0). Given ONLY the guess (Yes or No) and probability, no other words or explanation. For example: \n Guess: <most likely guess, as short as possible; not a complete sentence, just the guess!>\n Probability: <the probability between 0.0 and 1.0 that your guess is correct, without any extra commentary whatsoever; just the probability! \n \n The first QA pair is: \n Q: $\{THE QUESTION\} \n A: $\{THE ANSWER\} \n The second QA pair is: \n Q: $\{THE QUESTION\} \n A: $\{THE ANSWER\} |

Table 7: Prompt templates for different objectives, including semantically equivalent perturbations of input questions and semantic equivalent check of QA pairs.

to optimally balance efficiency (light version) and accuracy (performance version), facilitating more informed decisions in practice.

## C.2 Semantic Consistency Checking

The goal of our research is to provide a flexible framework for the effective detection of hallucinations in black-box LLMs. Compared to previous approaches (Manakul et al., 2023) based on similarity metrics such as BERTScore, employing LLM for semantic consistency checking not only offers better accuracy but also eliminates the need for involving additional models. We note that our design is not dependent on any specific type of LLM and the users are allowed to choose freely for each component. Specifically, we chose GPT as the instantiation in our experiments due to its well-acclaimed ability to follow human instructions which is cru-

cial for evaluation. In practice, the user may choose any open-sourced LLMs that have been aligned to follow instructions.

## C.3 Data Annotations

For the two classification QA tasks (Prime Number and Senator Search), we use the synthesized hallucinated answers following prior work (Zhang et al., 2023a). For the task of prime number, the factual/true answer is always "Yes", i.e., all the testing numbers are prime numbers. The synthesized hallucinated response is "No". The factual/true answer of the senator search is always "No" and the synthesized hallucinated answer is "Yes". Such an experimental setting on the binary classification tasks is realistic since we have verified that on these datasets most of the responses generated by gpt-3.5-turbo are indeed "Yes" or "No" which

align with the synthesized response.

For the generation QA tasks (HotpotQA-halu and NQ-Open-halu), we used answers generated by LLM (`gpt-3.5-turbo`) for experiments, which do not have pre-defined factual/non-factual labels. Therefore, we manually annotated these LLM-generated answers by comparing them with the ground truth.

We would like to note that in practice, a more versatile hallucination detection approach should be able to evaluate the factuality of a sample regardless of its origin (e.g., synthesized or generated by itself / other LLMs). In our framework, this is achieved through semantically equivalent question perturbation and cross-model response consistency checking. We plan to release the annotated datasets to facilitate future research.

## D  Additional Examples

Table 6 provides another illustrative example to explain our methodology in the senator search dataset.