# OpenReview forum: "SAC$^3$: Reliable Hallucination Detection in Black-Box Language Models via Semantic-aware Cross-check Consistency"
_EMNLP/2023/Conference — EMNLP 2023 Findings_

### Official Review · Reviewer_vZxH · 2023-08-03

**Soundness:** 3

**Excitement:**

4: Strong: This paper deepens the understanding of some phenomenon or lowers the barriers to an existing research direction.

**Paper Topic And Main Contributions:**

This paper presents a method called SAC^3, which aims to detect hallucinations using black-box LLMs. The proposed method starts by paraphrasing a question into multiple semantically equivalent questions. It then checks the consistency of the answers provided by the LLM to both the original question and the paraphrased questions to determine if the answers are factual. Additionally, this paper suggests incorporating the answers from other LLMs in the consistency checking process. The proposed method demonstrates better performance compared to the existing self-consistency approach on four datasets. The paper also investigates the impact of hyperparameters such as the number of paraphrased questions and the choice of threshold value.

**Questions For The Authors:**

- I believe that the current evaluations only consider synthesized datasets where the non-factual and factual answers are predetermined. This means you manually determine whether the answers are correct and the ratio of correct and incorrect answers. This could potentially create a gap between the results of the current evaluation and the actual behavior of LLMs in real scenarios. Therefore, could you further evaluate the methods using real outputs from LLMs on randomly sampled questions. Please correct me if I have misunderstood anything.

**Reasons To Accept:**

- A new method based on question paraphrase and cross-model checking is proposed for hallucination detection.
- The proposed method outperforms self-consistency by a large margin.

**Reasons To Reject:**

- The proposed method is computationally costly because it uses LLMs in multiple steps, such as paraphrasing the question, verifying semantic equivalency, obtaining the answer, and checking consistency.
- The evaluation all on synthesized datasets with pre-defined non-factual and factual answers, instead of real LLMs outputs.

**Reproducibility:**

4: Could mostly reproduce the results, but there may be some variation because of sample variance or minor variations in their interpretation of the protocol or method.

**Reviewer Confidence:**

3: Pretty sure, but there's a chance I missed something. Although I have a good feel for this area in general, I did not carefully check the paper's details, e.g., the math, experimental design, or novelty.

---

> ### Author Rebuttal · Authors · 2023-08-27
>
> Dear Reviewer vZxH,
>
> Thank you very much for your thoughtful comments. We are glad that you found the proposed method new and results promising (outperforms self-consistency by a large margin). Please find our point-by-point responses to your questions below. We are committed to fully addressing your concerns and are happy to provide additional clarification should you have any remaining questions.
>
> >**W1: The proposed method is computationally costly because it uses LLMs in multiple steps, such as paraphrasing the question, verifying semantic equivalency, obtaining the answer, and checking consistency.**
>
> Thanks for the comment. High computational cost is a common research challenge faced in many areas of black-box LLM studies, including self-consistency [1] and self-check [2]. Compared with these existing works, our framework enables model-level parallel execution of LLM inferences (either target or verifier LM) in the form of parallel API calls, which helps to mitigate the overhead. Moreover, we may further employ more advanced prompt strategies for semantic consistency checking that condense the pairwise comparisons to a single inference call to substantially reduce the time complexity. Finally, our framework also offers the flexibility for the user to adjust the balance between computational cost and accuracy: users may choose to trade off accuracy for efficiency in scenarios where low overhead is more valued than high detection precision. We will clarify this.
>
> >**W2&Q: I believe that the current evaluations only consider synthesized datasets where the non-factual and factual answers are predetermined. This means you manually determine whether the answers are correct and the ratio of correct and incorrect answers. This could potentially create a gap between the results of the current evaluation and the actual behavior of LLMs in real scenarios. Therefore, could you further evaluate the methods using real outputs from LLMs on randomly sampled questions. Please correct me if I have misunderstood anything.**
>
> We would like to clarify that our evaluations considered both synthesized outputs and real LLMs outputs.
>
> For the generation QA tasks (HotpotQA-halu and NQ-open-halu), we used answers generated by LLM (``gpt-3.5-turbo``) for experiments, which do not have pre-defined factual/non-factual labels. Therefore, we manually annotated these LLM-generated answers by comparing them with the ground truth (Section 5, lines 401-413).
>
> For the two classification QA tasks (Prime Number and Senator Search), we use the synthesized hallucinated answers following prior work [3]. For the task of prime number, the factual/true answer is always "Yes", i.e., all the testing numbers are prime numbers. The synthesized hallucinated response is "No". The factual/true answer of the senator search is always "No" and the synthesized hallucinated answer is "Yes". Such an experimental setting on the binary classification tasks is realistic since we have verified that on these datasets most of the responses generated by ``gpt-3.5-turbo`` are indeed "Yes" or "No" which align with the synthesized response.
>
>
> **References**
>
> [1] Wang, Xuezhi, et al. "Self-Consistency Improves Chain of Thought Reasoning in Language Models." The Eleventh International Conference on Learning Representations. 2022.
>
> [2] Manakul, Potsawee, Adian Liusie, and Mark JF Gales. "Selfcheckgpt: Zero-resource black-box hallucination detection for generative large language models." arXiv preprint arXiv:2303.08896 (2023).
>
> [3] Zhang, Muru, Ofir Press, William Merrill, Alisa Liu, and Noah A. Smith. "How language model hallucinations can snowball." arXiv preprint arXiv:2305.13534 (2023).

---

### Official Review · Reviewer_eydx · 2023-08-05

**Soundness:** 4

**Excitement:**

3: Ambivalent: It has merits (e.g., it reports state-of-the-art results, the idea is nice), but there are key weaknesses (e.g., it describes incremental work), and it can significantly benefit from another round of revision. However, I won't object to accepting it if my co-reviewers champion it.

**Paper Topic And Main Contributions:**

The paper focuses on the phenomenon of hallucinations in LLMs, and proposes a novel approach to verify the factuality of responses, that improves the self-consistency strategy. These improvements include: question-level improvements, where the question is replaced with semantically equivalent variants, which are generated and verified by an LLM (via designated prompts); and model-level improvements, where a verifier LLM is employed to also answer the question. In each of these improvements, the consistency of the answer is observed, namely across the different variants of the question and across the different models (and also the combination of question variants and models) to identify inconsistencies that could point to potential hallucinations.

**Questions For The Authors:**

In Table 1 in the Cross-Q/M variants, the authors considered positive answers with different explanations as non-consistent. This does not align with the original self-consistency work, that stated that the consistency of the answered is determined by the final answer of the model, and not the CoT part of which, which in my opinion is somewhat equivalent to the “explanation part” of the exemplified answers. Could the authors please elaborate on why these answers were considered inconsistent?

**Reasons To Accept:**

1.	This paper proposes a novel approach for identifying hallucinated responses, and for generating more trust-worthy responses.
2.	The authors conduct very thorough analyses, that demonstrate the efficacy of their approach on the gpt-3.5-turbo model.

**Reasons To Reject:**

1. The authors tested their strategy solely on the gpt-3.5-turbo model. Though their results seemed promising, I believe it is important to test their strategy on additional models, including at open-sourced models, to get a better picture of the strategy’s efficacy. Currently, with just one model being tested, this feels insufficient to fully assess the helpfulness of the author’s proposal.
2. The paper mostly dealt with "prompt-engineering", which is not that novel. Despite the potential merits of the proposed prompting strategy, I believe this alone might not be sufficiently innovative. Perhaps with more analyses and perturbations, this work could have had a noticeable contribution.

**Reproducibility:**

5: Could easily reproduce the results.

**Reviewer Confidence:**

5: Positive that my evaluation is correct. I read the paper very carefully and I am very familiar with related work.

---

> ### Author Rebuttal · Authors · 2023-08-27
>
> Dear Reviewer eydx
>
> Thank you very much for your detailed review and valuable suggestions for improving our manuscript. We are glad to hear that you found our idea novel and our analysis thorough. Please find our point-by-point responses to your questions below. We are committed to fully addressing your concerns and are happy to provide additional clarification should you have any remaining questions.
>
> >**W1: The authors tested their strategy solely on the gpt-3.5-turbo model. Though their results seemed promising, I believe it is important to test their strategy on additional models, including at open-sourced models, to get a better picture of the strategy’s efficacy. Currently, with just one model being tested, this feels insufficient to fully assess the helpfulness of the author’s proposal.**
>
> Our proposed framework does not restrict the type of LLM employed and can be naturally extended to various types of target LLMs. To verify this, in addition to GPT-3.5 (``gpt-3.5-turbo``), we have conducted additional experiments using GPT-4 (``gpt-4``) and PaLM 2 (``chat-bison``) on the considered four datasets. The experimental results of comparing the proposed SAC$^3$ with the SC$^2$ baseline are summarized in the following table. We observe that the proposed SAC$^3$ consistently outperforms the SC$^2$ baseline across all LLM variants.
>
> |       Methods           | Prime Number | Senator Search | HotpotQA-halu | NQ-open-halu |
> |-----------------------|:--------------:|:--------------:|:-------------:|:------------:|
> | SC$^2$ (GPT-3.5)    | 48.2         |      29.6      |      74.2     |     70.5     |
> | SC$^2$ (GPT-4)      | 38.3         |      18.4      |      79.7     |     76.3     |
> | SC$^2$ (PaLM 2)     | 26.9         |      19.2      |      75.8     |     67.9     |
> | SAC$^3$-Q (GPT-3.5) | 93.2         |      97.0      |      81.3     |     77.2     |
> | SAC$^3$-Q (GPT-4)   | 91.1         |      61.6      |      87.2     |     82.9     |
> | SAC$^3$-Q (PaLM 2)  | 90.3         |      66.3      |      82.8     |     72.7     |
>
>
> >**W2: The paper mostly dealt with "prompt-engineering", which is not that novel. Despite the potential merits of the proposed prompting strategy, I believe this alone might not be sufficiently innovative. Perhaps with more analyses and perturbations, this work could have had a noticeable contribution.**
>
> We would like to emphasize that our major contribution is to propose a novel and unified framework, SAC$^3$, that utilizes sampling-based semantic-aware cross-check consistency for effective hallucination detection in black-box LMs. SAC$^3$ is built upon a series of designs, including question perturbation, model perturbation, and semantic cross-check consistency. It is the combination of these design elements that enabled SAC$^3$ to achieve substantial improvements over the self-consistency baseline by a considerable margin (as high as 13.8%). The proposed prompting strategy is only one of the key components and is designed in place of the previous similarity-based consistency-checking techniques that failed to accurately assess semantic equivalency.
>
> >**Q: In Table 1 in the Cross-Q/M variants, the authors considered positive answers with different explanations as non-consistent. This does not align with the original self-consistency work, that stated that the consistency of the answered is determined by the final answer of the model, and not the CoT part of which, which in my opinion is somewhat equivalent to the “explanation part” of the exemplified answers. Could the authors please elaborate on why these answers were considered inconsistent?**
>
> In this work, we considered the task of hallucination detection provided with only the (potentially hallucinated) target response and without CoT or explanation. Our approach achieves this through semantically equivalent question perturbation and cross-model response consistency checking.
>
> The illustrative examples shown in Table 1 are regarding the input question ``Is 3691 a prime number?`` and the target response is ``No, 3691 is not a prime number. It is divisible by 7 and 13``, as mentioned in the caption.  We define the target question-answering (QA) pair by combining the input questions and the target response. The Cross-Q/M variants present a set of sampled QA pairs including perturbed questions and the corresponding answers. The semantic consistency between the target QA pair and sampled QA pairs:
>
> - Target QA pair:
> ``Q: Is 3691 a prime number? A: No, 3691 is not a prime number. It is divisible by 7 and 13``
>
>
> - Sampled QA pair 1:
> ``Q: Is the number 3691 prime? A: Yes``
>
> - Sampled QA pair 2:
> ``Q: Are the only factors of 3691 itself and one? A: Yes, the only factors of 3691 are 1 and itself.``
>
> - Sampled QA pair 3:
> ``Q: Can 3691 only be divided by 1 and 3691? A: Yes, 3691 can only be divided by 1 and 3691.``
>
> The target QA pair is non-factual since 3691 is in fact a prime number, while the three sampled QA pairs are not semantically consistent with the target QA pair as they are factual.

---

### Official Review · Reviewer_G3kV · 2023-08-11

**Soundness:** 3

**Excitement:**

3: Ambivalent: It has merits (e.g., it reports state-of-the-art results, the idea is nice), but there are key weaknesses (e.g., it describes incremental work), and it can significantly benefit from another round of revision. However, I won't object to accepting it if my co-reviewers champion it.

**Paper Topic And Main Contributions:**

This paper introduces a sampling-based hallucination detection method that addresses the limitations of traditional self-consistency-based detection methods. The specific contributions are as follows:
1. The discovery of two types of hallucination that are challenging to detect through self-consistency: question-level hallucination and model-level hallucination.
2. For these two types of hallucination, the authors propose SAC^3, an approach that calculates a weighted hallucination score through Question-level Cross-checking and Model-level Cross-checking.
3. Significant performance improvements are achieved compared to self-consistency method on four datasets in the tasks of classification QA and generation QA.

**Questions For The Authors:**

1. Line 527, should it be "and the verifier LM is a domain specific model developed for this domain"?
2. For the collection of the two classification QA datasets, the hallucinated samples are synthesized or gpt-3.5-turbo generated?
3. Will the annotated dataset open-source if this paper is accepted?

**Reasons To Accept:**

1. This paper is well written.
2. The proposed method can serve as a complementary approach to extend self-consistency methods.
3. Exceptionally strong results were achieved on the two datasets for the classification QA task, with AUROC values approaching 1. Additionally, in the generation QA task, the method also outperformed the baseline, showcasing improved performance.

**Reasons To Reject:**

1. The method requires a substantial number of API calls/LLM inference, encompassing question rewriting, response sampling, and semantic consistency checking, which is cost heavy.
2. The experimental results demonstrate remarkable performance gains of the method over the baseline in the classification QA task. However, the improvement in the generation QA task is not as pronounced. The authors should discuss the potential reasons for this phenomenon in the experimental section.
3. It lacks experiments on whether the proposed method can be applicable to various types of LLMs as the target model other than gpt-3.5-turbo.
4. It is unclear whether it is possible to employ open-source LLMs for semantic consistency checking, and whether the resulting performance decrease fall within an acceptable range.
5. The method proposed by the authors also encounters the same challenge as self-consistency, as depicted on the right side of Figure 1: even though the responses exhibit inconsistency among themselves, the response to be detected is actually factual.

**Reproducibility:**

4: Could mostly reproduce the results, but there may be some variation because of sample variance or minor variations in their interpretation of the protocol or method.

**Reviewer Confidence:**

4: Quite sure. I tried to check the important points carefully. It's unlikely, though conceivable, that I missed something that should affect my ratings.

---

> ### Author Rebuttal · Authors · 2023-08-27
>
> Dear Reviewer G3kV
>
> Thank you very much for the constructive comments and feedback. We are encouraged to hear that you found our paper well-written and our results exceptionally strong. Please find our point-by-point responses to your questions below. We are committed to fully addressing your concerns and are happy to provide additional clarification should you have any remaining questions.
>
> > **W1: The method requires a substantial number of API calls/LLM inference, encompassing question rewriting, response sampling, and semantic consistency checking, which is cost heavy.**
>
> Thanks for the comment. High computational cost is a common research challenge faced in many areas of black-box LLM studies, including self-consistency [1] and self-check [2]. Compared with these existing works, our framework enables model-level parallel execution of LLM inferences (either target or verifier LM) in the form of parallel API calls, which helps to mitigate the overhead. Moreover, we may further employ more advanced prompt strategies for semantic consistency checking that condense the pairwise comparisons to a single inference call to substantially reduce the time complexity. Finally, our framework also offers the flexibility for the user to adjust the balance between computational cost and accuracy: users may choose to trade off accuracy for efficiency in scenarios where low overhead is more valued than high detection precision. We will clarify this.
>
> > **W2: The experimental results demonstrate remarkable performance gains of the method over the baseline in the classification QA task. However, the improvement in the generation QA task is not as pronounced. The authors should discuss the potential reasons for this phenomenon in the experimental section.**
>
> Thank you for your observation. Our method shows superior performance on both tasks, while the improvement of classification QA is more significant. This is because the considered two classification QA tasks (i.e., Prime number and Senator search) are purposely crafted by prior work [3] to capture the worst scenario where the target LM such as GPT-3.5 is highly inclined to hallucinate and produces consistently wrong responses, which leaves more room for improvements. In contrast, the considered open-domain generation QA tasks are taken from general-purpose datasets that represent the average-case scenario. Although detecting hallucinations in the generative tasks is more challenging, our method still outperforms the baseline by a significant margin, namely achieving 13.8\% and 6.7\% improvement compared with the self-consistency baseline. We will add corresponding discussions.
>
> > **W3: It lacks experiments on whether the proposed method can be applicable to various types of LLMs as the target model other than gpt-3.5-turbo.**
>
> Our proposed framework does not restrict the type of LLM employed and can be naturally extended to various types of target LLMs. To verify this, in addition to GPT-3.5 (``gpt-3.5-turbo``), we have conducted additional experiments using GPT-4 (``gpt-4``) and PaLM 2 (``chat-bison``) on the considered four datasets. The experimental results of comparing the proposed SAC$^3$ with the SC$^2$ baseline are summarized in the following table. We observe that the proposed SAC$^3$ consistently outperforms the SC$^2$ baseline across all LLM variants.
>
> |       Methods           | Prime Number | Senator Search | HotpotQA-halu | NQ-open-halu |
> |-----------------------|:--------------:|:--------------:|:-------------:|:------------:|
> | SC$^2$ (GPT-3.5)    | 48.2         |      29.6      |      74.2     |     70.5     |
> | SC$^2$ (GPT-4)      | 38.3         |      18.4      |      79.7     |     76.3     |
> | SC$^2$ (PaLM 2)     | 26.9         |      19.2      |      75.8     |     67.9     |
> | SAC$^3$-Q (GPT-3.5) | 93.2         |      97.0      |      81.3     |     77.2     |
> | SAC$^3$-Q (GPT-4)   | 91.1         |      61.6      |      87.2     |     82.9     |
> | SAC$^3$-Q (PaLM 2)  | 90.3         |      66.3      |      82.8     |     72.7     |
>
> > **W4: It is unclear whether it is possible to employ open-source LLMs for semantic consistency checking, and whether the resulting performance decrease fall within an acceptable range.**
>
> The goal of our research is to provide a flexible framework for the effective detection of hallucinations in black-box LLMs. Compared to previous approaches [2] based on similarity metrics such as BERTScore, employing LLM for semantic consistency checking not only offers better accuracy but also eliminates the need for involving additional models. We note that our design is not dependent on any specific type of LLM and the users are allowed to choose freely for each component. Specifically, we chose GPT as the instantiation in our experiments due to its well-acclaimed ability to follow human instructions which is crucial for evaluation. In practice, the user may choose any open-sourced LLMs that have been aligned to follow instructions.
>
> > **W5: The method proposed by the authors also encounters the same challenge as self-consistency, as depicted on the right side of Figure 1: even though the responses exhibit inconsistency among themselves, the response to be detected is actually factual.**
>
> Thanks for the comment. We agree with the reviewer on that precisely detecting non-factual responses through inconsistency remains an open research problem and is difficult to address fully. However, compared to the existing approach that solely relies on model self-consistency, our proposed approach aims to mitigate this by weighing in the responses generated by multiple LMs with question perturbations, which, as verified by our extensive experiments, is more reliable and robust against various types of hallucinations and significantly outperforms the self-check baselines.
>
> >**Q1: Line 527, should it be "and the verifier LM is a domain specific model developed for this domain"?**
>
> Thanks for your careful reading. We will correct this in our final revision.
>
> > **Q2: For the collection of the two classification QA datasets, the hallucinated samples are synthesized or gpt-3.5-turbo generated?**
>
> For the two classification QA tasks (Prime Number and Senator Search), we use the synthesized hallucinated answers following prior work [3]. For the task of prime number, the factual/true answer is always "Yes", i.e., all the testing numbers are prime numbers. The synthesized hallucinated response is "No". The factual/true answer of the senator search is always "No" and the synthesized hallucinated answer is "Yes". Such an experimental setting on the binary classification tasks is realistic since we have verified that on these datasets most of the responses generated by ``gpt-3.5-turbo`` are indeed "Yes" or "No" which align with the synthesized response.
>
> For the generation QA tasks (HotpotQA-halu and NQ-open-halu), we used answers generated by LLM (``gpt-3.5-turbo``) for experiments, which do not have pre-defined factual/non-factual labels. Therefore, we manually annotated these LLM-generated answers by comparing them with the ground truth (Section 5, lines 401-413).
>
> We would like to note that in practice, a more versatile hallucination detection approach should be able to evaluate the factuality of a sample regardless of its origin (e.g., synthesized or generated by itself / other LLMs). In our framework, this is achieved through semantically equivalent question perturbation and cross-model response consistency checking.
>
> >**Q3: Will the annotated dataset open-source if this paper is accepted?**
>
> Yes, we plan to release all the source code and annotated datasets (e.g., JSON files) for better reproducibility and also facilitate future research.
>
> **References**
>
> [1] Wang, Xuezhi, et al. "Self-Consistency Improves Chain of Thought Reasoning in Language Models." The Eleventh International Conference on Learning Representations. 2022.
>
> [2] Manakul, Potsawee, Adian Liusie, and Mark JF Gales. "Selfcheckgpt: Zero-resource black-box hallucination detection for generative large language models." arXiv preprint arXiv:2303.08896 (2023).
>
> [3] Zhang, Muru, Ofir Press, William Merrill, Alisa Liu, and Noah A. Smith. "How language model hallucinations can snowball." arXiv preprint arXiv:2305.13534 (2023).

---

### Meta-Review · Area_Chair_bMym · 2023-09-19

**Recommendation:** 3

**Metareview:**

This paper proposes to address two types of hallucinations due to question-level and model-level that are beyond the existing self-consistency detection method. Reviewers appreciate that this is a nice contribution to hallucination detection which demonstrates strong performances. The author also provides further results on GPT-4 and PaLM 2 in their rebuttal to show it can consistently deliver improvements. In addition to these new results, we strongly encourage the authors to take the feedback to improve the paper over computational costs, its usefulness for open-source LLMs, and the unresolved challenge of detecting non-factual responses through inconsistency.

---

### Decision · Program_Chairs · 2023-10-07

**Decision:**

Accept-Findings

**Comment:**

This paper proposes to address two types of hallucinations due to question-level and model-level that are beyond the existing self-consistency detection method. Reviewers appreciate that this is a nice contribution to hallucination detection which demonstrates strong performances. The author also provides further results on GPT-4 and PaLM 2 in their rebuttal to show it can consistently deliver improvements. In addition to these new results, we strongly encourage the authors to take the feedback to improve the paper over computational costs, its usefulness for open-source LLMs, and the unresolved challenge of detecting non-factual responses through inconsistency.